# Heart rate variability before and after 14 weeks of training in Thoroughbred horses and Standardbred trotters with different training experience

Zsófia Nyerges-Bohák[☾], Krisztina Nagy[‡], László Rózsa[‡], Péter Póti[‡], Levente Kovács[ID]*[☾]

Institute of Animal Sciences, Hungarian University of Agriculture and Life Sciences, Kaposvár, Hungary

☾ These authors contributed equally to this work.
‡ These authors also contributed equally to this work.
* Kovacs.Levente@uni-mate.hu

**Data Availability Statement:** The data used in this study are available at the Harvard Dataverse repository at doi:10.7910/DVN/NP14UA (https://

## Abstract

Changes in heart rate and heart rate variabilty (**HRV**) were investigated in untrained (UT; starting their first racing season) and detrained (DT; with 1–3 years of race experience) racehorses before and after 14-week conventional training. HRV was measured at rest over 1 h between 9:00 and 10:00 AM on the usual rest day of the horses. The smallest worthwhile change (**SWC**) rate was calculated for all HRV parameters. UT horses had significantly higher heart rate compared to DT ($P<0.001$). There were no gender- or training-related differences in heart rate. The root-mean-square of successive differences (**rMSSD**) in the consecutive inter-beat-intervals obtained after the 14-week training period was lower compared to pre-training rMSSD ($P<0.001$). The rMSSD was not influenced by breed, age or gender. In DT horses, there was a significant decrease in the high frequency (**HF**) component of HRV ($P\leq0.05$) as the result of the 14-week training. These results may reflect saturation of high-frequency oscillations of inter-beat intervals rather than the reduction in parasympathetic influence on the heart. The HF did not differ significantly between the two measurements in UT horses; however, 16.6% of the animals showed a decrease in HF below SWC ($P\leq0.05$). This supports the likelihood of parasympathetic saturation. Although no significant decrease in heart rate was found for the post-training, 30.0% of DT and 58.3% of UT horses still showed a decrease in heart rate below the SWC. Also by individual examination, it was also visible that despite significant post-training decrease in rMSSD, 1 (4.6%) DT and 2 (6.7%) UT horses reached SWC increase in rMMSD. In the case of these horses, the possibility of maladaptation should be considered. The present results indicate that similar to as found in human athletes, cardiac ANS status of racehorses also changes during the physiological adaptation to training. To explore more precise links between HRV and training effectiveness in horses, a more frequent recording would be necessary. Detailed analysis of HRV parameters based on SWC will be able to highlight the importance of fitness evaluation at individual level.

dataverse.harvard.edu/dataset.xhtml?persistentId=
doi:10.7910/DVN/NP14UA).

**Funding:** The New National Excellence Program of
the Ministry of Human Capacities supported this
study in the form of funding to ZN-B [ÚNKP-17-3-
IV-ÁTE-3]. This study was also supported by the
OTKA Research Scholarship of the National
Research, Development and Innovation Office in
the form of funds to LK [Budapest, Hungary; K-
134204]. Hylofit provided sensors for ZN-B for the
HRV measurements. The funders had no role in
study design, data collection and analysis, decision
to publish, or preparation of the manuscript.

**Competing interests:** The authors have read the
journal's policy and have the following competing
interests: Hylofit provided sensors for Zsófia
Nyerges-Bohak for the HRV measurements. There
are no patents, products in development or
marketed products associated with this research to
declare. This does not alter our adherence to PLOS
ONE policies on sharing data and materials.

## Introduction

Assessment of performance and fitness has been conducted in racehorses for many years and the methods available have become progressively more refined [1]. However, these approaches (i.e., precise placement of a heart rate monitor and the frequent check for proper signal transmission during riding) are time-consuming for the work riders/drivers and the related serial blood sampling makes them inconvenient in everyday workouts.

Heart rate (**HR**; the number of heartbeats per unit of time) and heart rate variability (**HRV**; the short-term fluctuations in the successive cardiac interbeat intervals) have increasingly been used for the assessment of the parasympathetic nervous system (**PNS**) activity in human athletes from the last decades [2–4]. Time domain measures of HRV, especially the root mean square of successive differences (**rMSSD**) between the consecutive interbeat-intervals (**IBI**) and the high frequency (**HF**) component of HRV were found to be strongly associated with PNS activity in horses as well [5]. The PNS activity has been described as a marker correlated to performance [6–8] and may also be useful to design and control the recovery periods [9–11]. Regular assessment of HRV at rest can help to set optimal training loads by tracking the training adaptation/maladaptation in human athlets [12,13].

In horses, HRV analysis is frequently used for stress [14–16], pain [17,18], and behavioral investigations [19,20]; however, the sport physiological implications of HRV is a lesser-studied area. Most of the equine sports physiology studies examined cardiac autonomic responses during exercise [21–27], and the majority of the papers concluded that HRV analysis is not useful for the evaluation of ANS activity in the heavily exercising horse. The long-term relationship between HRV parameters measured at rest and fitness was rarely studied in horses; in addition, the results are also contradictory. Some authors suggest that parasympathetic activity may be increased as a result of a training program [28], while others found that the parasympathetic dominance can be fully activated even in untrained horses [29]. Similar to as found in human athletes [6,30,31], Kinnunen et al. [32] observed no linear relationship between HRV and fitness in trotters.

The purpose of present study was to investigate the effect of 14-week training on HRV recorded at rest in racehorses. To recognize the effects of conditions besides fitness, horses of different gender, age and breed were involved to the trial. The establishment of a non-invasive, inexpensive, time-efficient technique, capable of conveying athletic training status would be an irreplaceable innovation in equestrian sports. The present investigation aims to be the first step toward this goal.

## Materials and methods

### Horses

The experiment was specifically approved by the Ethics Committee for Animal Experiments of the University of Veterinary Medicine (University of Veterinary Medicine Budapest, H-1078 Budapest, István utca 2., Hungary). The study was approved by the Pest County Government Office, Department of Animal Health (Permit Number: PE/EA/1973-6/2016). Sixty-eight horses were enrolled into the study (**Table 1**). Horses starting their first racing season were considered as untrained (**UT**) group. Horses with previous training experience (1–2 years) were considered as detrained (**DT**), because all of them had at least 3 month rest period before trial start. All of the UT horses were two, while DT horses were 3 or 4 years old at the time of the experiment. Horses were kept in individual loose boxes (3.0 ×3.0 m) on straw and were fed three times a day with hay and concentrates (oat, oat balancer concentrated racehorse mix, bran). Water and mineral supplements were always available. Seven horses were excluded

**Table 1. The number of the horses participated in the study according to training experience, breed and gender.**

|  | Thoroughbreds (n = 30) | | Standardbred trotters (n = 24) | |
|---|---|---|---|---|
|  | Untrained (n = 17) | Detrained (n = 13) | Untrained (n = 7) | Detrained (n = 17) |
| **Female** | 11 | 7 | 6 | 5 |
| **Male** | 6 | 6 | 1 | 12 |

from the analysis as they were unexpectedly sold, injured or became ill during the trial and seven individuals were excluded because of poor-quality HRV recordings (dried out electrodes, poor skin-electrode contact, interrupted IBI signal transmission between the sensor and the receiver). None of the remaining horses showed signs of overreaching/overtraining, lameness or other health problems during the study period. In total, data of 30 Thoroughbred horses and 24 Standardbred trotters were used for the study before and after the 14-week standard training (**Table 2**).

All horses performed a 6-week pre-training period to build a foundation of fitness through trotting work, hill work and, in case of Thoroughbreds, the initial canter work. After pre-training HRV recordings, experimental horses started preparing for races with a more intensive pace work period. Horses were trained routinely with a gradually increasing intensity for 14 weeks by the same trainer. The quantification of training load was defined based on external indicators of effort intensity, i.e. distance, approximate velocity, and number of repetitions. Characteristics of the workout schedule are summarized in **Table 2**. The training schedule for the UT and DT horses basically did not differ; however, the exercise intensity was adjusted by a qualified trainer to the individual abilities and development of an individual horse. Speed and distance of the training was increased gradually throughout the 14 weeks for all horses based on the assessment of the trainer.

### Experimental setting and data recording

Measurements were carried out in April 2017 (pre-trainnig recordings) and in August 2017 (post-training recordings). Post-training recordings took place between 2 and 4 days after the completion of the 14-week training period. Since ANS activity is highly sensitive to the previous day activity [10,32], horses performed only mild exercise during the last two days before HRV recordings. Based on recommendations for farm animals [5] time periods when horses

**Table 2. The weekly training schedules of the experimentalhorses.**

|  | Thoroughbreds | Standardbreds |
|---|---|---|
| Monday | Paddock; for horses which raced Mild canter;for the others | Intensive trotting |
| Tuesday | Intensive gallop | Mild trotting |
| Wednesday | Mild canter | Intensive trotting or RACE |
| Thursday | Mild canter | Rest (Paddock) |
| Friday | Intensive gallop | Mild trotting |
| Saturday | Mild canter | Rest or RACE |
| Sunday | Rest or RACE | Paddock; for horses which raced Mild trotting; for others |

Mild canter: Warming up + 2–3000 m canter (8–10 m/s); Intensive gallop: Warming up + 1500–3000 m submaximal exercise (11–13 m/s) (gradual increase in the distance and velocity during the 12 week); Mild trotting: Warming up + 8–9000 m trotting (6–7 m/s); Intensive trotting: Warming up + 1500–3000 m submaximal exercise (10–11 m/s) (gradual increase in the distance and velocity during the 12 week).

were calm and undisturbed (without excitement associated with feeding and other morning stable activities) were used for the analysis. Inter-beat-intervals (**IBI**) were recorded over 1 h between 9:00 and 10:00 AM on the usual rest day of the horses with a Polar Equine RS800CX multi device and a Polar H2 sensor (Polar Ltd., New York, USA) [33]. Two hours were given for the animals after feeding to get accustomed with the equipment before the start of the data recording.

All horses followed the routine training schedule adjusted to its abilities during the 14 weeks between the trial sessions. They all raced at least one time in the season, but none of the horses participated at race in the last two weeks before the post-training recording.

## HRV analysis

Four equal length of 5-min continuous IBI samples were selected from the 1-h recording for all horses either for the pre- and post-training recordings based on visual inspection of the IBI tachogram. The Kubios HRV software (version 2.2, Biomedical Signal Analysis Group, Department of Applied Physics, University of Kuopio, Finland) was used for the analysis of IBI data. Artifact correction was made as in previous studies on horses [15,25] following established procedures [34]. During the frequency domain analysis of the IBI data, the fast Fourier transformation was applied to calculate the high-frequency (**HF**) component of HRV, which represents parasympathetic activity [5]. The frequency range of HF was set up from 0.13 to 0.26 Hz according to von Borell et al. [5]. In the time domain, the HR (1/min) and the root mean square of the successive differences (**rMSSD**) in the consecutive IBIs (ms) were calculated. The rMSSD is the primary time domain measure used to estimate vagal regulatory activity in large animals [5].

## Statistical analysis

For hypothesis testing, a general linear mixed model (**GLMM**) was fit to the data [35] with random effects for each horse. The fixed effects were training experience (UT vs. DT), gender (mare vs. stallion), breed (Standardbred vs. Throughbred) test session (pre-training vs. post-training) and their interaction. The HR, rMSSD, and HF were inserted into the models as response variables. Logarithmic transformation of HR and rMSSD parameters was applied to satisfy the normality and variance homogeneity assumptions of the models. For multiple comparisons the Tukey-Kramer correction was used.

To avoid incorrect conclusion when assessing changes in a HRV parameter, the smallest worthwhile change (**SWC**) was also identified [36,37]. SWC is a commonly used parameter in human sport science, since the ability to reliably detect meaningful changes in performance tests over time is of great importance [38,39]. The SWC of HR, rMSSD and HF were calculated for UT and DT horses separately [40]. The prevalence of horses successfully achieving SWC by pre- and post-training HRV recordings were compared between UT and DT group using the Fisher test.

The significance level was set at $P \leq 0.05$ in cases of all tests. All analyses were carried out using the R 4.0.3. statistical software (R Core Team R Vienna, Austria) [41].

## Results

Amongst the examined two breeds (Thoroughbred horses and Standardbred trotters) the ratio of mares (45.9% vs. 60%, $P = 0.411$) and the ratio of UT horses (29.2% vs. 56.7%, $P = 0.057$) did not differ significantly. The HR of mares was higher ($P \leq 0.05$) compared to stallions/geldings. UT horses had significantly higher HR compared to DT ones ($P \leq 0.001$). 30.0% of DT

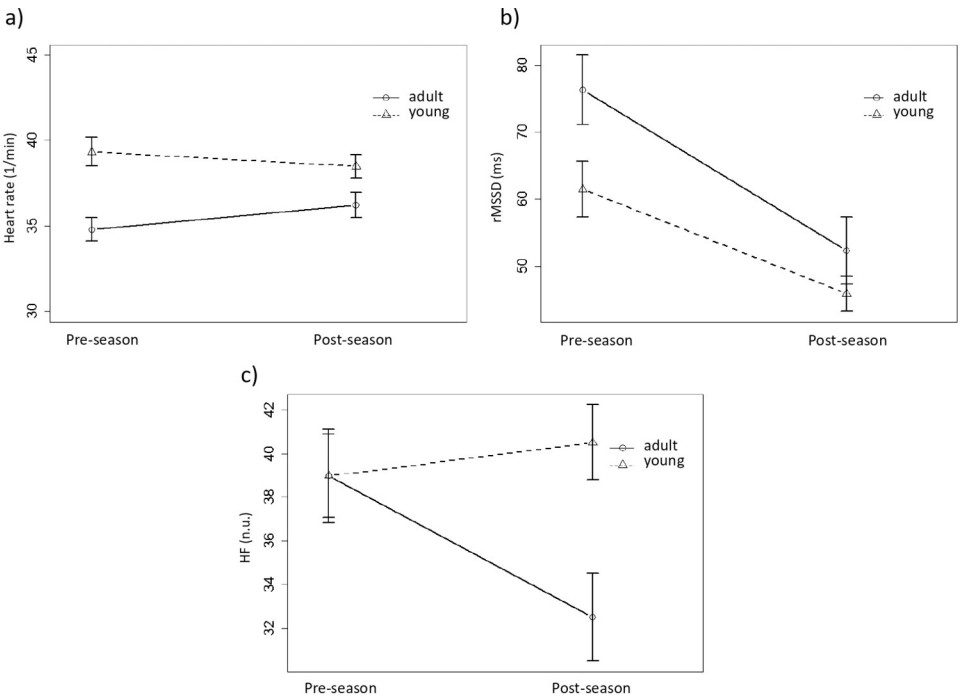

**Fig 1.** Heart rate (a), rMSSD (b) and HF (c) before (pre-training) and after (post-training) a 14-week conventional training in racehorses. Values are presented as means ± SEM. rMSSD = root-mean-square of successive differences in the consecutive inter-beat-intervals. HF = the high frequency component of heart rate variability.

horses and 58.3% of UT horses showed a decrease in HR below SWC between the two measurements; however, this difference was not significant ($P = 0.053$) (**Fig 1**).

Data of rMSSD obtained post-training was significantly lower compared to pre-training data ($P \leq 0.001$) (**Fig 1**). RMSSD was not influenced significantly by breed, group or gender. Only 1 (4.6%) DT horse and 2 (6.7%) UT horses reached the SWC increase in regards to rMMSD, trainng experience did not has a significant effect.

UT horses had higher HR compared to DT horses ($P < 0.001$) recorded either pre- or post-training. Moreover the HF was affected not only by the 14-week training ($P \leq 0.05$) but a training × group interaction was also found ($P \leq 0.05$) (**Fig 1**). The HF parameter was affected by time ($P \leq 0.05$) and it's interaction with training experience ($P \leq 0.05$) (**Fig 1**). 16.6% of UT horses showed a decrease in HF below SWC ($P \leq 0.05$) at the post-training measurements.

## Discussion

Although the HR of mares was higher compared to stallions ($P \leq 0.05$), there was no significant HRV difference according to gender. Gender-related HR difference probably do not reflect the magnitude of modulation in parasympathetic outflow in horses, it may be rather the result of the significantly smaller left ventricles of mares [42]. Gender seems to be significant influencing factor of HRV in human athletes [43] but results in horses are controversial. Although some studies found no gender-related differences in HR [19,20,32], some authors observed differences between mares and stallions [25,28,44]. These differences may have occurred due to differing training, handling and stabling protocols between females and males [25,44] or limitations in data recording [28]. In the latter study, the intensity of the previous day's workload was not taken into account, which possibly resulted in the reduced vagal tone found in mares on the day of recordings [32].

A significant decrease in HF from pre-training to post-training values was observed in DT horses, but not in UT individuals. All UT horses were from 1 to 2 years younger than DT horses, thus it arises that training × group interaction might influenced by age.

Age-related changes in HR and HRV have been frequently reported in horses [26,28,45,46], but according to these studies, these are relevant findings in the case of a larger age difference. Because the age difference between the two group was only 1–2 years, our results reflect more likely the different training experience as opposed to aging per se.

In the human sport science, HF is an appropriate measure in predicting athletic performance [11], while in racehorses its physiological meaning is less well understood [28,29]. Training induced decrease in HF found by DT horses is still a surprising outcome, because effective training is generally thought to be associated with increases in vagal-related indices of HRV in humans [9,47,48], while reductions in vagal tone have been related to fatigue and/or non-functional overreaching [49–51]. This controversial result is not limited to DT horses, because decreased rMSSD from the pre-training measurements to the post-training session was found in both groups. Low rMSSD has been showed as reliable marker of fatigue and poor recovery in endurance horses immediately after exercise [26]; however, exercise-induced long term changes in rMSSD has not been investigated in horses so far.

Most of the investigations on human athletes suggest that high HRV is associated with high fitness, whereas low HRV corresponds with low fitness; however, these studies tested non-professional athletes [9,47,48]. In elite athletes with extensive training histories or when training loads approach higher levels, the HRV response to training tend to be markedly different; in similar longitudinal studies of elite athletes, decreases in HRV were linked to increses in fitness [6,52]. This suggests that reductions in HRV are not neccesarily associated with fatigue [50,51,53] but may reflect positive adaptation to the training. Horses are generally accepted as extraordinary natural athletes among mammals [54], moreover, horses used in this study followed the standard Hungarian racehorse training plan. The course of their adaptation to training is more similar to that of elite human athletes than to recreational individuals.This reduced HRV even with effective training can be explained with a physiological phenomenon, that has not yet been studied in horses; it is the so-called parasympathetic saturation. According to Plewset al. [31] increased vagal tone may give rise to sustained parasympathetic control of the sinus node, which may eliminate respiratory heart modulation and reduce HRV. This phenomenon may have manifested itself in present experiment, and the effect seems to be more pronounced in DT horses with lower post-training HF, caused by the more extensive training experience. The decreased HF below SWC at the post-training measurements in UT horses supports that some of the UT horses also tended to show more explicit signs of parasympathetic saturation. In the case of such saturation in human athletes, the vagal-related ANS indices are substantially reduced, while HR decreases [30]. Although we found no significant HR differences between pre- and post-training recordings, when examined individually, 30.0% of DT and 58.3% of UT horses showed a decrease in HR below SWC, which further increases the likelihood of parasympathetic saturation. Only one horse from the DT group and two horses from the UT group showed (the previously expected) SWC increase in regards to rMMSD. In the case of these horses, the possibility of maladaptation should be considered. A series of reports on humans highlight the importance of non-significant changes in sport if their magnitude is actually greater than the SWC [40]. It might be worthwhile to incorporate this precision approach into horse sports physiology as well. It should be also mentioned, that course of vagal activity throughout a training process was observed to be bell-shaped in several studies [6,30,31]. Cardiac autonomic regulation improves during the initial phase of training, while it decreases over the weeks preceding competition [55]. This is in line with some results found in

racehorses, where the average HRV was higher in the middle of the training process than before races [32], moreover, precompetition stress further reduced HRV in trotters [16].

## Conclusion

Cardiac ANS status of horses changed significantly after 14 weeks of training. The assessment of HRV seems to be a promising approach for determination of training status and for individualizing the training program in racehorses. However, to draw definitive conclusions about the relationships between HRV and the expectable athletic achievements, further research with larger sample sizes and more frequent recordings are needed. The work out induced parasympathetic saturation should be also further investigated in racehorses. Future studies would help researchers to work out successful HRV-guided training methods for equestrian sport professionals.

## Acknowledgments

The authors thank Bettina Viczena for her assistance with this work.

## Author Contributions

**Conceptualization:** László Rózsa, Levente Kovács.

**Methodology:** Levente Kovács.

**Resources:** Péter Póti.

**Software:** Krisztina Nagy.

**Supervision:** Péter Póti.

**Validation:** Krisztina Nagy, Levente Kovács.

**Writing – original draft:** Zsófia Nyerges-Bohák.

**Writing – review & editing:** László Rózsa, Levente Kovács.

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
