## [Decision Letter · Decision Letter 0]

18 Aug 2021

PONE-D-21-19666

Heart rate variability before and after a race season in Thoroughbred horses and Standardbred trotters with different training experience

PLOS ONE

Dear Dr. Kovács,

Thank you for submitting your manuscript to PLOS ONE. After careful consideration, we feel that it has merit but does not fully meet PLOS ONE’s publication criteria as it currently stands. Therefore, we invite you to submit a revised version of the manuscript that addresses the points raised during the review process.

We look forward to receiving your revised manuscript.

Kind regards,

Chris Rogers

Academic Editor

PLOS ONE

2. In your Methods section, please provide additional details regarding the horses used in your study and ensure you have described the source. For more information regarding PLOS' policy on materials sharing and reporting, see https://journals.plos.org/plosone/s/materials-and-software-sharing#loc-sharing-materials.

“The authors thank Bettina Viczena for her assistance with this work. The research was supported by Hylofit. Zsófia Nyerges-Bohak was supported by the New National Excellence Program of the Ministry of Human Capacities [ÚNKP-17-3-IV-ÁTE-3].”

“The funders had no role in study design, data collection and analysis, decision to publish, or preparation of the manuscript”

Additional Editor Comments (if provided):

Both reviewers have made some useful comments about points that require clarification. Please look at these points and address these.

Reviewers' comments:

Reviewer's Responses to Questions

**Comments to the Author**

1. Is the manuscript technically sound, and do the data support the conclusions?

Reviewer #1: No

Reviewer #2: Partly

2. Has the statistical analysis been performed appropriately and rigorously? 

Reviewer #1: Yes

Reviewer #2: I Don't Know

3. Have the authors made all data underlying the findings in their manuscript fully available?

Reviewer #1: No

Reviewer #2: Yes

4. Is the manuscript presented in an intelligible fashion and written in standard English?

Reviewer #1: Yes

Reviewer #2: Yes

5. Review Comments to the Author

Reviewer #1: Review of the manuscript No: PONE-D-21-19666

Manuscript Title: Heart rate variability before and after a race season in Thoroughbred horses and

Standardbred trotters with different training experience

ABSTRACT

Lines: 12-14: Names of the investigated groups should be changed – please, see the comment below (Mat. and methods).

Line 17: Abstract section is is not the right place to discuss the obtained results (“which may be the result of…”).

Line 20: Another interpretation of the results (“indicating decrease…”). Please present this as a conclusion from the study, if it is important. And please remember that the interpretation of the obtained results should be discussed in Discussion section only. I think the authors fell into a kind of trap because they did not separate the Results and Discussion sections in the main text of the manuscript. Therefore, in the Abstract section, they used a similar type of description that is not appropriate in this place.

Line 23 and 26: The next comment concerns the marking of statistical significance. It is generally recognized that if there is a significance of differences in results, we consider them as a P≤0.05 or P≤0.01. In my opinion, they are more readable than those in the text (e.g. P=0.030 or P=0.012).

Line 27: Another interpretation of the results.

Lines 31-32: Another interpretation of the results.

Line 32: This statement cannot be considered a research conclusion (please note that, in its current form it is only the authors' assumption).

It seems that in the Abstract there is too little information about the material on which the experiment was conducted and about the methods used. Hence, there are disproportionately many results presented and their interpretations (sic!). Lastly, there are no unequivocal conclusions, only the authors' assumptions (e.g. “The present result suggest...”). Please consider this form: “The present results indicate that…”.

INTRODUCTION

Line 41: What approaches do the authors mean? Please explain this statement or provide examples from the available literature.

Lines 44-46: First, HR and HRV are two completely separate indicators. Please do not describe them together because it may be mistakenly perceived by the readers. Secondly, if there are examples of HRV parameters, please provide more than one!

Lines 51-61. I cannot agree with the Authors at all. There are many studies on the effects of training on horse HRV parameters, e.g. Frick et al. doi: 10.1111/jvim.15358), Munsters et al. (doi: 10.1017/S1751731112002327), Younes et al. (doi: 10.3389/fphys.2016.00155), Christensen et al. (doi.org/10.1016/j.physbeh.2014.01.024), Kowalik et al. (doi: 10.1111/asj.12671). These studies are not more than 15-20 years old, so please remove this remark entirely from the work. I especially recommend reading the last mentioned work (doi: 10.1111/asj.12671), it contains a well-presented problem of research on HRV in racehorses. In view of the above, I propose to rewrite this whole paragraph, taking into account the latest worldwide research on HRV.

Lines 64-66: Once again, I disagree with the Authors - this method (HRV assessment) has been used for years and cannot be considered as innovative in equine sport medicine. Please carefully consider modifying the purpose of the research performed.

MATERIAL AND METHODS

Lines 71-72: Which institution approved the research? Please add the detailed name and seat of the University (city, country etc.).

Lines 74-76: In my opinion, the descriptions of the studied groups of horses are debatable. The Authors do not provide the real/actual age of the tested horses, but only describe their training experience, namely as young horses they consider horses without training experience, and as adults - horses with 1-3 years of experience. This is unacceptable even from a physiological point of view, since both horses (“young” and “adult”) can be adult in fact. Therefore, firstly, information about the real age of the tested horses in the groups should be added (in the text or as separate table), and secondly - I recommend changing the names of the groups to, for example, "untrained" and "trained".

Line 80: What was the potential reason for getting "poor quality HRV recordings"? Why were horses with this writing disorder completely excluded from experience? This statement suggests an error in the measuring apparatus, was it not possible to correct its settings?

Line 83: “after 12 weeks of training” What kind of training? Please refer to the legend to Figure 1 and standardize the type of training used in horses (was it standard training? If so - please write about it here).

Lines 90-93: It is somewhat incomprehensible that the horses identified as YOUNG had exactly the same training program as ADULT. This may suggest that, being untrained, they will have problems with keeping up with the horse's performance. Or vice versa, the ADULT horses did not have loads consistent with their high degree of training advancement. In such a situation, the division into these groups of horses is pointless. Please, explain it more precisely, especially since some training loads differences have been described above.

Paragraph 95-105: There is a lot of questions here. Since pre-training / seasonal recordings started in April, and post-training recordings in August, why do the Authors stated that the study lasted 12 weeks? In fact, from the last week of April to the first week of August there are 14 weeks. So, please precise it. My second question is why the Authors used the term “post-seasonal” recordings here? Is the training season really over in August? If not, maybe it would be better to use the terms "pre-training recordings and" post-training recordings "? Please, consider it.

Line 97: “…intensive training period.” – as mentioned above, after a careful analysis of the training schedule, doubts arise as to its really high intensity. On the basis of what parameters (e.g. LA blood level) the Authors assessed the training intensity in both groups of horses? Such information is necessary to confirm whether the training used was really intense. Please provide such data, if possible. If not, please do not use the phrase "intensive training period".

Next question concern the timing of IBI recording. Namely, the authors first note that for such tests, horses should rest without any normal morning activities (see lines: 100 & 101), and then describe that the measurement of IBI was taken after all these activities. Do these statements sound contradictory? In fact, the Authors later mention that the horses had 2 hours of rest, but did this time allow the animals to return to their emotional resting state? How was this confirmed?

Finally, the last comment on this paragraph concerns the term used in line 106. Namely, it is about "development" - in the context of the whole sentence it suggest that at the beginning of the study, some of the horses were undeveloped (somatically? too young? others reasons?). Maybe it would be better to change the word to “experience”, for example?

Lines 117-119: Please do not discuss your chosen method (HF) in this chapter! If in doubt, please discuss it in the Discussion section.

Line 123: why “in farm animals”? or only in farm animals? Can you please expand on this statement?

Line 126: “young vs. adult” – please read my earlier comment.

Line 127: “pre-season vs. post-season” – please consider to change the name the factors / groups to “pre-training” and “post-training”. I suggest this change because in different countries the training season covers usually a longer period (for example from March to October), depending on national regulations.

Lines 133-134: The description of SWC parameter is not necessary here – it should be given in Material and Methods, HRV analysis sub-section.

Line 136: as was mentioned before: not “for adult and young horses”, but for example “trained and untrained horses” .

Line 140: Please, give the full name of the statistical software, with the manufacturer name, and country of origin.

RESULTS AND DISCUSSION

First of all, in accordance to PLOS One Authors’ guidance, the “middle section” of the manuscript should consist of following sections: Materials and Methods, Results, Discussion, and Conclusions (optional). So please consider splitting the current Results and Discussion section. This will facilitate a substantive discussion of the obtained results and drawing appropriate conclusions. In the current version, the lack of such a division introduces some chaos and causes that the obtained results are practically unnoticeable.

Moreover, the combination of these subsections in one probably also meant that the authors did not provide any significant conclusions from the study. Namely, in the Conclusion section, the Authors only reported that 12 weeks of training changed the ANS status of the tested animals. The remaining content of this section seems to be the Authors' deliberations only.

Tables and Figures are clear.

References: All of 55 references are cited in the text, and all are given in accordance with the guidelines for the Authors.

Finally, I do not recommend this manuscript in current version. I hope that my comments will help to improve the text of the manuscript and it will be finally published in PLOS ONE.

Reviewer #2: This manuscript describes interesting information in the field of equine cardiology. However, I have some major concerns regarding study design, language and grammar. My main study design concern is only two measurements of HRV. It seems like a lot of things could affect the horses and that drawing conclusions from only 2 readings may be in appropriate. I was wondering if an ECG was analyzed from the polar or if this polar only recorded R-R intervals. There is no mention of what was done with any arrhythmias (second degree AV block, etc). Lastly, I feel unqualified to completely analyze the statistical analysis and ask that someone with a stronger statistical background review this section.

Line 18 Please put a space in between ‘inheart’

Line 32 Please put a space after ‘considered.’

Line 35 Please put a space in between ‘individuallevel’

Line 35-36 The conclusion is difficult to understand and does not fit with the conclusion in the rest of your paper where you discuss general patterns and findings. I agree that more information is needed but his sentence makes is sound like your study cannot provide any useful ‘generalizations’ in the quine sport science field.

Line 47 Please rewrite, ‘and may be also’ to say ‘and may also be’

Line 51 The word ‘popular’ is not correct. It’s often discussed but infrequently used in real life, consider re-wording this.

Line 56 The manuscript describes the last studies as being published 15-20 years ago. There are more frequent publications. See Lorello et al 2017 as an example and there may be a few more

Line 64 The manuscript discusses different breeds and only 2 different breeds were used. This is ok, but considering all of the horse breeds around, 2 different breeds is not a large collection.

Line 77 Please clarify what ‘mash’ is

Line 80 Place a space between ‘HRVrecordings’

Line 81 The manuscript describes no signs of overreaching/overtraining. These signs in horses are not to my knowledge well described. Can you elaborate or clarify what you mean?

Line 119 Place a space in between ‘rangeof’

Line 152 Pace a space in between ‘inthe’

Line 171 Correct spelling of ‘athletes’

Line 175 Correct spelling of ‘elite’ and ‘decreases’ and change ‘was’ to ‘were linked’

Line 181 Place a space between ‘explainedwith’

Line 213 Please rephrase ‘work out’

Table 2 Place a space in between ‘experimentalhorses’

6. PLOS authors have the option to publish the peer review history of their article (what does this mean?). If published, this will include your full peer review and any attached files.

Reviewer #1: No

Reviewer #2: No

---

## [Author Response · Author response to Decision Letter 0]

7 Oct 2021

'Response to Reviewers'

We would like to thank the Reviewers for the generous and constructive help in correction of the manuscript. We appreciate your time spent with the review. We feel that the Reviewers’comments and recommendations were reasonable, and we tried to take them into account as far as possible while improving the manuscript. The activities of both Reviewers contributed significantly to the improvement of the quality of our paper.

Reviewer1’s comments are addressed below.

ABSTRACT

Comment: Lines: 12-14: Names of the investigated groups should be changed – please, see the comment below (Mat. and methods). 

AU: Thank you for your suggestion. I agree it is more appropriate if the names of the groups refer to the fitness status of the horses and not the age of them. I changed the group names to untrained and detrained, which also emphasise that the formerly called adult group started the experiment after rest period. I did not give the precise age data of the horses, because I think not the real age, rather the training experience is relevant to the experiment. However, I have specified that untrained group includes 2-year-old beginner horses, and detrained group includes 3- or 4-year-old experienced racehorses.

Comment: Line 17: Abstract section is not the right place to discuss the obtained results (“which may be the result of…”).

AU: Line 16-17: Thank you, this part of the sentence has been deleted.

Comment: Line 20: Another interpretation of the results (“indicating decrease…”). Please present this as a conclusion from the study, if it is important. And please remember that the interpretation of the obtained results should be discussed in Discussion section only. I think the authors fell into a kind of trap because they did not separate the Results and Discussion sections in the main text of the manuscript. Therefore, in the Abstract section, they used a similar type of description that is not appropriate in this place. 

AU: For readers unfamiliar with HRV numerical values, we highlighted that decreased rMSSD means physiologically decreased vagal tone. We believed this might help in the interpretation of the abstract. However, we accepted your suggestion and deleted the interpretation of the results.

Comment: Line 23 and 26: The next comment concerns the marking of statistical significance. It is generally recognized that if there is a significance of differences in results, we consider them as a P≤0.05 or P≤0.01. In my opinion, they are more readable than those in the text (e.g., P=0.030 or P=0.012). + Line 27: Another interpretation of the results. + Lines 31-32: Another interpretation of the results.

AU: Thank you for your comments, the marking of statistical significance has been changed throughout the manuscript.

Comment: Line 32: This statement cannot be considered a research conclusion (please note that, in its current form it is only the authors' assumption).

It seems that in the Abstract there is too little information about the material on which the experiment was conducted and about the methods used. Hence, there are disproportionately many results presented and their interpretations (sic!). Lastly, there are no unequivocal conclusions, only the authors' assumptions (e.g. “The present result suggest...”). Please consider this form: “The present results indicate that…”.

AU: Lines 30-35: the preliminary nature of the result has been better highlighted in the abstract.

INTRODUCTION

Comment: Line 41: What approaches do the authors mean? Please explain this statement or provide examples from the available literature.

AU: It is the author’s personal experience that properly install and adjust the heart rate monitor is stressful for work riders during saddling (Line 41-42). Although these instruments are suitable for measuring horses, it takes at least an extra 5 minutes/ horse to achieve proper signal transmission during saddling. When measuring HRV at rest, these 5 minutes are also needed, but it takes place in horse’s rest period, and not during the pre-work stress and hurry.

Comment: Lines 44-46: First, HR and HRV are two completely separate indicators. Please do not describe them together because it may be mistakenly perceived by the readers. Secondly, if there are examples of HRV parameters, please provide more than one! 

AU: Lines 44-45: Thank you for your comment. Separate definitions are provided for the terms HR and HRV. Following your suggestion, we also provided examples for HRV parameters (rMSSD and HF; Lines 47-50).

Comment: Lines 51-61. I cannot agree with the Authors at all. There are many studies on the effects of training on horse HRV parameters, e.g. Frick et al. doi: 10.1111/jvim.15358), Munsters et al. (doi: 10.1017/S1751731112002327), Younes et al. (doi: 10.3389/fphys.2016.00155), Christensen et al. (doi.org/10.1016/j.physbeh.2014.01.024), Kowalik et al. (doi: 10.1111/asj.12671). These studies are not more than 15-20 years old, so please remove this remark entirely from the work. I especially recommend reading the last-mentioned work (doi: 10.1111/asj.12671), it contains a well-presented problem of research on HRV in racehorses. In view of the above, I propose to rewrite this whole paragraph, taking into account the latest worldwide research on HRV. + Lines 64-66: Once again, I disagree with the Authors - this method (HRV assessment) has been used for years and cannot be considered as innovative in equine sport medicine. Please carefully consider modifying the purpose of the research performed. 

AU: Lines 58-60: My feeling is that the message of this part of the article is misunderstood. The relationship between HRV and exercise stress is indeed not novel in the field of equestrian experiments either. However, as we have tried to highlight in the manuscript (presumably in the wrong way), most of the articles do not examine “general” resting HRV in terms of fitness, but are used to examine response to certain situations, recovery, or acute stress caused by the acute workout. This is also a useful area, but long term HRV changes caused by the physiological adaptation of the body and the consequent improvement of fitness is not described in horses, and is even an actively researched, constantly changing area in humans.

However, the problem is definitely with the first draft of the manuscript, if the paragraph was ambiguous to the Reviewer, I tried to rephrase it in a clearer way. 

I have already read the articles proposed in the review, but in my opinion, all of them look at HRV as an indicator of mental / physical stress or possible abnormal arrhythmias and not as an indicator of a horse’s fitness. 

Frick et al. doi: 10.1111/jvim.15358 investigated HRV during exercise. We mentioned in the manuscript, that “most of the equine sports physiology studies examined cardiac autonomic responses during exercise”. We also give 6 references for that; however, it is right, that Frick et al. is not listed among them. The reason is, that Frick did not examine the HRV per se, they tried to explore possible arrhythmias during treadmill exercise using the method HRV measurement.

Munsters et al. 2013 (doi: 10.1017/S1751731112002327) aimed to evaluate physiological and behavioral responses of police horses during police training. However, the procedure is called „training”, it does not mean fitness workout. Horses were trained to better withstand stressful situations and HRV was measured in various behavioral tests (horses encountered four challenging objects positioned at a distance of 20m or more from each other, etc.) In the manuscript we also mentioned: „In horses, HRV analysis is popular for stress [14–16], pain [17; 18], and behavioral investigations [19; 20]”, and we gave 7 references for that. 

Younes et al. doi: 10.3389/fphys.2016.00155 is the number [26]in the reference list.

Christensen et al. doi.org/10.1016/j.physbeh.2014.01.024 aimed to investigate acute stress responses of dressage horses ridden in three different Head-and-Neck-positions. This article is also one of the studies investigating the correlations between HRV and stress and pain.

Kowalik et al. 2017 (doi: 10.1111/asj.12671) have investigated young racehorses before and during warming up. They explored the beneficial effects of relaxation massage. They wrote in M&M section: “HRV was recorded for 10 min to obtain the resting HR and HRV levels. Recording was continued during 10min of grooming and saddling the horse and the next 10 min of warm-up walking.” As we can read, the resting values were recorded immediately before riding. As that, HRV expresses the horse’s emotional connection to riding, the so-called anticipatory reaction. (Bohák et al. 2018; DOI:10.1371/journal.pone.0201691) As the author emphasize, stress is a very important factor in horses’ performance, but not in fitness. Although Kowalik and collaborates’ article is very high quality and innovative, we miss that it doesn’t discuss the impact of back pain in young horses. Based on my personal experience, a large percentage of yearlings are struggling with chronic pain, which also affects their relationship to training and their performance. Massage is one of the best ways to prevent back muscle pain. For me, the result obtained suggests that saddling and the rider on their back did not cause discomfort to the treated horses, so the anticipatory stress was also lower. However, in my opinion, the article does not measure the true resting HRV values of horses and does not draw any conclusions about the relationship between resting HRV and fitness. 

MATERIAL AND METHODS

Comment: Lines 71-72: Which institution approved the research? Please add the detailed name and seat of the University (city, country etc.).

AU: Lines 75-76: We filled in the missing data.

Comment: Lines 74-76: In my opinion, the descriptions of the studied groups of horses are debatable. The Authors do not provide the real/actual age of the tested horses, but only describe their training experience, namely as young horses they consider horses without training experience, and as adults - horses with 1-3 years of experience. This is unacceptable even from a physiological point of view, since both horses (“young” and “adult”) can be adult in fact. Therefore, firstly, information about the real age of the tested horses in the groups should be added (in the text or as separate table), and secondly - I recommend changing the names of the groups to, for example, "untrained" and "trained".

AU: Lines 79-82. Thank you for your suggestion. Authors agree it is more appropriate if the names of the groups refer to the fitness status of the horses and not the age of them. I changed the group names to untrained and detrained, which also emphasise that the formerly called adult group started the experiment after rest period. We did not give the precise age data of the horses, because I think not the real age, rather the training experience is relevant to the experiment. However, we specified that the untrained group includes 2-year-old beginner horses, and detrained group includes 3- or 4-year-old experienced racehorses.

Comment: Line 80: What was the potential reason for getting "poor quality HRV recordings"? Why were horses with this writing disorder completely excluded from experience? This statement suggests an error in the measuring apparatus, was it not possible to correct its settings?

AU: Poor recording quality means that the horse-sensor contact or the sensor-Polar watch signal transmission contact was not correct (Lines 87-88), so the data set downloaded after the measurement was empty or contained signal error. Since the horses were not disturbed during the measurement, we were not able to check the proper signal transmission during the measurement.

Comment: Line 83: “after 12 weeks of training” What kind of training? Please refer to the legend to Figure 1 and standardize the type of training used in horses (was it a standard training? If so - please write about it here).

AU: Table 1 and Table 2 were swapped. Horses were trained followed a standard training program. Table 2 includes the detailed training schedule.

Comment: Lines 90-93: It is somewhat incomprehensible that the horses identified as YOUNG had exactly the same training program as ADULT. This may suggest that, being untrained, they will have problems with keeping up with the horse's performance. Or vice versa, the ADULT horses did not have loads consistent with their high degree of training advancement. In such a situation, the division into these groups of horsesis pointless. Please, explain it more precisely, especially since sometraining loads differences have been described above. 

AU: Lines 98-100: The given training program is a general schedule where wementioned an interval in terms of distances and paces in the horse training program. The training of the horses wasadjusted by a qualified trainer to the individual abilities and development of the horse. As that the untrained and the detrained group worked following the similar schedule but not with the similar intensity. The goal was to expose all horses to the same internal load, which of course means a slightly different external load for each individual.

Comment: Paragraph 95-105: There is a lot of questions here. Since pre-training / seasonal recordings started in April, and post-training recordings in August, why do the Authors state that the study lasted 12 weeks? In fact, from the last week of April to the first week of August there are 14 weeks. So, please precise it. My second question is why the Authors used the term “post-seasonal” recordings here? Is the training season really over in August? If not, maybe it would be better to use the terms "pre-training recordings and" post-training recordings "? Please, consider it. 

AU: Both suggestions are legitimate, we apologize for the inattention, we have corrected the inaccuracy.

Comment: Line 97: “…intensive training period.” –as mentioned above, after a careful analysis of the training schedule, doubts arise as to its really high intensity. On the basis of what parameters (e.g. LA blood level) the Authors assessed the training intensity in both groups of horses?Such information is necessary to confirm whether the training used was really intense. Please provide such data, if possible. If not, please do not use the phrase "intensive training period".

AU: Line 94: It is a known limitation of the study, that we did not have the possibility to check the training intensity with LA blood level, etc. However, horses followed the standard Hungarian training schedule (Line 203) according to the instructions of a qualified and experienced trainer. Nevertheless, the phrase "intensive training period" has been changed, which only suggested that this was the real training period following the pre-training foundation.

Comment: Next question concern the timing of IBI recording. Namely, the authors first note that for such tests, horses should rest without any normal morning activities (see lines: 100 & 101), and then describe that the measurement of IBI was taken after all these activities. Do these statements sound contradictory? In fact, the Authors later mention that the horses had 2 hours of rest, but did this time allow the animals to return to their emotional resting state? How was this confirmed?

AU: We can only eliminate distractions as much as possible. The quietest time between the two feedings should be chosen for the measurement. As preparations for the next feeding begin at 11 a.m. (which means further excitement for the horses), we have indicated this period as the optimal measurement time as possible.

Comment: Finally, the last comment on this paragraph concerns the term used in line 106. Namely, it is about "development" - in the context of the whole sentence it suggests that at the beginning of the study, some of the horses were undeveloped (somatically? too young? others reasons?). Maybe it would be better to change the word to “experience”, for example?

AU: Line 115: we were referring to training development with the term, but to avoid ambiguity, we replaced the term with "adjusted to its abilities".

Comment: Lines 117-119: Please do not discuss your chosen method (HF) in this chapter! If in doubt, please discuss it in the Discussion section.

AU: Lines 185-186: The marked explanation was moved to the discussion.

Comment: Line 123: why “in farm animals”? or only in farm animals? Can you please expand on this statement?

AU: Line 130: There are several time domain parameters, and these are not always preferred to use in all species. RMSSD is the primary time domain measure in large animals. It was corrected.

Comment: Line 126: “young vs. adult” – please read my earlier comment+ Comment: Line 136: as was mentioned before: not “for adult and young horses”, but for example “trained and untrained horses”.

AU: Line 133 (and in all sentences where appropriate) we have replaced these terms with "untrained and detrained."

Comment: Line 127: “pre-season vs. post-season” – please consider to change the name the factors / groups to “pre-training” and “post-training”. I suggest this change because in different countries the training season covers usually a longer period (for example from March to October), depending on national regulations.

AU: Lines 134-135 (and in all sentences where appropriate) we have replaced these terms with "pre-training and post-training."

Comment: Lines 133-134: The description of SWC parameter is not necessary here – it should be given in Material and Methods, HRV analysis sub-section. 

AU: Line 138: Although we partly agree with the Reviewer, for the sake of clarity, I still consider it reasonable to leave this one-line explanation here. SWC is a very rarely used indicator in veterinary science, and it is easier for the reader to interpret the article if he understands what the term SWC refers to.

Comment: Line 140: Please, give the full name of the statistical software, with the manufacturer name, and country of origin.

AU: Line 147: It has been corrected.

RESULTS AND DISCUSSION

Comment: First of all, in accordance to PLOS One Authors’ guidance, the “middle section” of the manuscript should consist of following sections: Materials and Methods, Results, Discussion, and Conclusions (optional). So please consider splitting the current Results and Discussion section. This will facilitate a substantive discussion of the obtained results and drawing appropriate conclusions. In the current version, the lack of such a division introduces some chaos and causes that the obtained results are practically unnoticeable.

Moreover, the combination of these subsections in one probably also meant that the authors did not provide any significant conclusions from the study. Namely, in the Conclusion section, the Authors only reported that 12 weeks of training changed the ANS status of the tested animals.The remaining content of this section seems to be the Authors' deliberations only. 

AU: Although at first it seemed easier to write the two chapters together, finally we agree with the Reviewer that it improves the quality of the manuscript if Results and Discussion are separated.

Reviewer2’s comments are addressed below 

ABSTRACT

Comment: Line 18 Please put a space in between ‘inheart’+ Line 32 Please put a space after ‘considered.’ + Line 35 Please put a space in between ‘individuallevel’+ Line 80 Place a space between ‘HRVrecordings’+ Line 119 Place a space in between ‘rangeof’ + Line 152 Pace a space in between ‘inthe’ + Line 181 Place a space between ‘explainedwith’+ Table 2 Place a space in between ‘experimentalhorses’

AU: Due to an unknown IT error, Microsoft Word takes some space characters out of the text during transmission. We will do our best to eliminate this problem.

Comment: Line 35-36 The conclusion is difficult to understand and does not fit with the conclusion in the rest of your paper where you discuss general patterns and findings. I agree that more information is needed but his sentence makes is sound like your study cannot provide any useful ‘generalizations’ in the equine sport science field.

AU: Lines 30-35: We highlighted the most important result, that – as in human beings - the HR and HRV is influenced by the training in horses.

INTRODUCTION

Comment: Line 47 Please rewrite, ‘and may be also’ to say ‘and may also be’

AU: Line 51: It has been corrected, thank you for your correction.

Comment: Line 51 The word ‘popular’ is not correct. It’s often discussed but infrequently used in real life, consider re-wording this.

AU: Line 54: The phrase has been changed to “frequently used”.

Comment: Line 56 The manuscript describes the last studies as being published 15-20 years ago. There are more frequent publications. See Lorello et al 2017 as an example and there may be a few more. 

AU: Line 58-60: My feeling is that the message of this part of the article is misunderstood. The relationship between HRV and exercise stress is indeed not new in the field of equestrian experiments either. However, as I have tried to highlight in the manuscript (presumably in the wrong way), most of the articles do not examine “general” resting HRV in terms of fitness, but are used to examine response to certain situations, recovery, or acute stress caused by the acute workout. This is also a useful area, but long term HRV changes caused by the physiological adaptation of the body and the consequent improvement of fitness is not described in horses, and is even an actively researched, constantly changing area in humans.

However, the error is definitely in the manuscript, if the paragraph was ambiguous to the Reviewer, I tried to rephrase it in a clearer way.

I have already read the article proposed in the review (Lorello et al 2017), but in my opinion, it reflects the acute stress response to exercise in eventing horses (recovery HRV), and not the long-term resting HRV changes due to the improved fitness. Recovery HRV also refers to fitness, but there are many influencing factor (different environmental condition, transport after the standard exercise test, etc.) which make this method less sensitive than measuring the normal resting HRV in the box.

MATERIAL AND METHODS

Comment: Line 64 The manuscript discusses different breeds and only 2 different breeds were used. This is ok, but considering all of the horse breeds around, 2 different breeds is not a large collection.

AU: This is right, but in the present case we had the possibility to examine the most common breeds in the two most common form of horse racing.

Comment: Line 77 Please clarify what ‘mash’ is

AU: Lines 83-84: Mash was concentrated racehorse mix + bran + water. We corrected it in the manuscript.

Comment: Line 81 The manuscript describes no signs of overreaching/overtraining. These signs in horses are not to my knowledge well described. Can you elaborate or clarify what you mean?

AU: The most common signs of overtraining is signs of fatigue and poor performance combined with weight loss, inappetence and signs of psychic stress including tachycardia, nervousness, muscle tremor, sweating and diarrhoea. None of these occurred in experimental horses.

RESULTS AND DISCUSSION

Comment: Line 171 Correct spelling of ‘athletes’+ Line 175 Correct spelling of ‘elite’ and ‘decreases’ and change ‘was’ to ‘were linked’ + Line 213 Please rephrase ‘work out’

AU: Lines 195; Line 199; Line 234: We accepted all the advice, thank you.

---

## [Editor Report · Decision Letter 1]

2 Nov 2021

Heart rate variability before and after 14 weeks of training in Thoroughbred horses and Standardbred trotters with different training experience

PONE-D-21-19666R1

Dear Dr. Kovács,

We’re pleased to inform you that your manuscript has been judged scientifically suitable for publication and will be formally accepted for publication once it meets all outstanding technical requirements.

Kind regards,

Chris Rogers

Academic Editor

PLOS ONE

Additional Editor Comments (optional):

Thank you for the edits to he manuscript. It is now suitable to progress with the publication process.
---

## [Editor Report · Acceptance letter]

1 Dec 2021

PONE-D-21-19666R1 

Heart rate variability before and after 14 weeks of training in Thoroughbred horses and Standardbred trotters with different training experience 

Dear Dr. Kovács:

I'm pleased to inform you that your manuscript has been deemed suitable for publication in PLOS ONE. Congratulations! Your manuscript is now with our production department. 

Kind regards, 

on behalf of

Dr. Chris Rogers 

Academic Editor

PLOS ONE